# Benefits of Polydopamine as Particle/Matrix Interface in Polylactide/PD-BaSO_4_ Scaffolds

**DOI:** 10.3390/ijms21155480

**Published:** 2020-07-31

**Authors:** Naroa Sadaba, Aitor Larrañaga, Gemma Orpella-Aceret, Ana F. Bettencourt, Victor Martin, Manus Biggs, Isabel A. C. Ribeiro, Jone M. Ugartemendia, Jose-Ramon Sarasua, Ester Zuza

**Affiliations:** 1Department of Mining-Metallurgy Engineering and Materials Science, School of Engineering EIB 1, University of the Basque Country (UPV/EHU) and Polymat, 48013 Bilbao, Spain; naroasadaba89@gmail.com (N.S.); aitor.larranagae@ehu.eus (A.L.); jone.munoz@ehu.eus (J.M.U.); jr.sarasua@ehu.eus (J.-R.S.); 2Center for Research in Medical Devices (CÚRAM), National University of Ireland (NUIG), Newcastle Road, H91 W2TY Galway, Ireland; gemma.orpella@gmail.com (G.O.-A.); manus.biggs@nuigalway.ie (M.B.); 3Research Institute for Medicines (iMed.ULisboa), Faculdade de Farmácia, Universidade de Lisboa, Avenida Prof. Gama Pinto, 1649-003 Lisboa, Portugal; asimao@ff.ulisboa.pt (A.F.B.); ribeiroi@campus.ul.pt (I.A.C.R.); 4Laboratory for Bone Metabolism and Regeneration, Faculty of Dental Medicine, U. Porto Rua Dr. Manuel Pereira da Silva, 4200-393 Porto, Portugal; victorzmartin@gmail.com; 5Portugal/LAQV/REQUIMTE, U. Porto, 4160-007 Porto, Portugal

**Keywords:** biodegradable, composite, polylactide, barium sulfate, polydopamine, melt processing, template, 3D-printing, scaffolds, antibiotic, levofloxacin

## Abstract

This work reports the versatility of polydopamine (PD) when applied as a particle coating in a composite of polylactide (PLA). Polydopamine was observed to increase the particle–matrix interface strength and facilitate the adsorption of drugs to the material surface. Here, barium sulfate radiopaque particles were functionalized with polydopamine and integrated into a polylactide matrix, leading to the formulation of a biodegradable and X-ray opaque material with enhanced mechanical properties. Polydopamine functionalized barium sulfate particles also facilitated the adsorption and release of the antibiotic levofloxacin. Analysis of the antibacterial capacity of these composites and the metabolic activity and proliferation of human dermal fibroblasts in vitro demonstrated that these materials are non-cytotoxic and can be 3D printed to formulate complex biocompatible materials for bone fixation devices.

## 1. Introduction

Polylactides are biodegradable polymers with great potential for the reconstruction of damaged tissues [1]. As is well known polylactide can be either semicrystalline (poly-L-lactide and poly-D-lactide (PLLA/PDLA)) or fully amorphous (poly-D,L-lactide (PDLLA)). For reconstruction of hard tissue stiff and strong PLLA or PDLA is preferred. Having a glass transition temperature around 60 °C, at body temperature semicrystalline PLA will be in glassy state (vs. elastomeric) and being crystalline at the same it will fulfil the requirements of a high stiffness and strength that a polymeric biomaterial requires for bone reconstruction in the form of fixation devices of small size [2]. Shortcomings to the use of PLA in bone reconstruction as opposed to e.g., metals (e.g., stainless steels or titanium alloys), include transparency to X-ray and mechanical brittleness [3,4]. However, high mass element containing fillers may be added to PLA to obtain radiopaque composites and improve its mechanical properties. Barium sulfate (BaSO_4_) [5,6,7], ferrous oxide (Fe_3_O_4_) [8] and bismuth oxide [9,10] have been previously reported as radiopaque composite additives.

Inorganic particulate reinforcements can enhance the mechanical properties of polymers and confer additional filler-specific properties to the matrix [11]. It is also well known that the enhancement of strength related mechanical properties (elastic modulus, resistance to yield and ultimate stress) is much more noticeable with continuous fiber composites [12], however continuous fiber formulations do not lend themselves well to free-form manufacturing techniques such as extrusion, injection molding or, as is the case, 3D printing [13,14,15]. Particularly, 3D printing is progressing very rapidly as an advanced manufacturing technique also for particulate composites. In this domain, there are also considerable achievements in the development of drug delivery systems. Therefore, in this work we selected a 3D printable system based on a BaSO_4_ particulate and PLA composites with an antimicrobial drug incorporation for the potential use in fixation devices for bone tissue regeneration.

Polydopamine is a newfangled nature inspired adhesive normally used for immobilization on materials surfaces of small molecules as drugs and proteins for biomedical use [16]. In addition, it has been used for improving the thermal stability of composites [17]. In most cases promoted by its biomimetic adhesivity [18], polydopamine has been used to confer anchoring sites for biologically active molecules as drugs and proteins, as nanocapsules or nanocarriers [19,20]. Nevertheless, the main use of polydopamine is as substrate coating due to its ability and efficiency for conjugation with bioactive molecules as growth factors [21] and drugs [22]. Zhang et al. immobilize bone morphogenic protein 2 (BMP-2) and insulin-like growth factor 1 (IGF-1) on polydopamine coated scaffold, reducing the burst release of the factors and endowing long-term osteoconductivity [21]. Furthermore, titanium nanotubes have been coated with polydopamine to adsorb dexamethasone sodium phosphate, an anti-inflammatory and modulator of osteogenic differentiation, leading to slow, sustained and controlled drug release [22]. However, few publications have reported the use of polydopamine in polymer matrix composites reinforced with inorganic and radiopaque compounds. Almost all the publications referred to composites focus their research on dispersion and biocompatibility, for example, on boron nitride nanotubes [23], bioglass in polylactides [17] and multi-walled carbon nanotubes (MWCNT) in polyurethanes (PU) [24]. Few works have reported changes in mechanical properties of polymer composites due to polydopamine coating of the inorganic phase, one of them is that on polylactide reinforced by polydopamine functionalized halloysite nanotubes [25].

The aim of this work is to study a novel polymeric biodegradable composite system having potential for use as a biodegradable polymeric fixation template applicable to bone reconstruction. In this sense a novel composite system is studied and developed including features of radiopacity (RO) [26], mechanical toughness [3,5,11,12] and antimicrobial properties for prevention of biofilm formation [27]. In this aim, barium sulfate particles were functionalized with polydopamine and incorporated into a PLA matrix to obtain scaffolds by 3D-printing. It will be proved that this composites presents X-ray visibility and enhanced mechanical properties (stiffness, strength, ductility and toughness) through the promotion of new deformation mechanisms caused by proper particle size and specific interactions at the fiber/matrix interfaces leading to good adhesion and improved mechanical performance. The incorporation of polydopamine to the systems presents additional benefits since provides specific sites for biologically active molecules that can be incorporated to the polymeric scaffolds leading to improvements in biocompatibility and other requirements for specific tissue engineering in situ. 

## 2. Results and Discussion

### 2.1. Improvements in the Mechanical Properties by Incorporation of Polydopamine Coated BaSO_4_ Particles into PLA

The tensile stress–strain curves and mechanical properties of neat polylactide and its composites (PLA/PD-BaSO_4_) were determined by tensile testing. Neat PLA, not reaching a yield point, showed typical brittle behavior, exhibiting around 6% of elongation before failure and 60 MPa of tensile strength. In contrast, PLA/PD-BaSO_4_ composites showed, for all PD-BaSO_4_ amounts studied, a clear yield point and an extended ductile behavior with a dramatic increase in elongation at break (Appendix A). This is accompanied with a moderate increase in the elastic modulus. As identified in our previous works with PLA/BaSO_4_ composites [3,5], the plastic deformation in composites occurs because the rigid particles act as stress concentrators and, after particles debonding from the matrix and being of a 0.7–1.9 µm size [28], activate, at a point of applied stress, crazing and shear yielding deformation mechanisms. Since polydopamine covered particles (PD-BaSO_4_) present 1.25 µm size and interactions with the matrix through the polydopamine interface, in these composites, increases in all mechanical properties (elastic modulus, strength, ductility and toughness) are achieved [29].

Table 1 shows the tensile mechanical properties of PLA/PD-BaSO_4_ composites. As can be observed, up to a 2 wt.% BaSO_4_ content in composites, Young’s modulus, ductility and toughness improvements can be observed. Further, if these results are to be compared with those of PLA/BaSO_4_, the non PD functionalized composite counterparts, enhancements in stiffness and strength together with ductility and toughness can be noticed in the former. Please also note that a huge >2300% increase in elongation at break are determined in these novel composite formulations in regard to neat PLA, which brings about a dramatic improvement in toughness. This is attributable to specific interactions stablished between the ester groups of polylactide and the alcohol groups of polydopamine coating of the particles surface [18,30] bringing about a stronger fiber/matrix interface. It is also noticeable that a 30% increase in the elongation at break is obtained in PD coated BaSO_4_ PLA composites in regard to the composites without polydopamine coating [3]. Finally, beyond the 2 wt.% particle composition, a slight general decrease in mechanical properties are noticeable, suggesting the existence of particle aggregates and bundles beyond this point.

Figure 1 represents the break surface of the PLA/PD-BaSO_4_ 2 wt.% composite. The SEM image corroborates the ductile and tough behavior of the PLA containing 2 wt.% barium sulfate PD-coated particles with values provided in Table 1. A high level of dispersion of PD coated particles (indicated with white and straight arrows) within the PLA matrix can be observed. In the pictures, ductile-deformed threads of PLA matrix appear too, thinner than a micrometer in diameter. This means that the matrix has been stretched during the tensile test beyond the yield point leaving long fibers (red dot-line arrows). Therefore, radiopaque PD-BaSO_4_ particles were able to act as fixed points that under the external applied stress assisting the PLA matrix to develop specific ductile deformation mechanisms.

It is also remarkable that the values obtained for the 2 wt.% and 10 wt.% composites do not differ much from each other. However, radiopacity values, as expected, show a significant increase with the increasing amount of PD-BaSO_4_ in composites (Appendix A) and particularly in the 10 wt.% composite. Consequently, it was concluded that among all compositions studied the 10 wt.% PLA/PD-BaSO_4_ composite presents the optimal properties (Appendix A) and hence an additional analysis is conducted for this composite.

The mechanical properties of the 3D printed scaffolds are also analyzed. Details of the scaffolds geometry design can be seen in Appendix A. In this case, mechanical properties were measured in compression to mimic the working conditions of the device. Non-reinforced polylactide (as reference) and the 10 wt.% PLA/PD-BaSO_4_ scaffolds with 55% of porosity have been tested.

Figure 2 shows the stress–strain curve’s behavior under compression of neat PLA scaffolds and those of its 10 wt.% PLA/PD-BaSO_4_ composite counterpart. As can be observed the 3D printed PLA and PLA/PD-BaSO_4_ scaffolds do show different regimes and each regime corresponds to a specific mechanism of a porous structured material, in agreement with bibliography. In the first stage (region I), the walls contribute to the resistance of the scaffolds under compressive load, which results in an elastic response region at initial loads and corresponding strains. In the second stage (II), the pores collapse by buckling of the walls (barreling effect). The third stage (III) is featured by a large increase in stress over strain which may be explained by the fact that scaffolds are now compressed to a size that the scaffold becomes denser and furthermore more strain resistant to the applied load [31,32,33]. Despite the fact that both scaffolds have the same porosity (55%), in Figure 2 the neat PLA behaved like a more rigid structure [34], while PLA/PD-BaSO_4_ shows again a flexible and softer behavior. Therefore, it can be concluded that the particles confer flexibility to the scaffolds in compression tests.

### 2.2. Biocompatibility Assessment

In vitro compatibility studies were performed to determine the possible toxicity of BaSO_4_ and PD-BaSO_4_ particles and their resulting PLA composites using Human dermal fibroblasts (HDFs) as a basic toxicity test. Figure 3 shows the metabolic activity of Human dermal fibroblasts (HDFs) in the presence of 10, 50, 100 or 500 µg/mL of BaSO_4_ or PD-BaSO_4_ particles. The metabolic activity displayed in this figure was normalized at each time-point with respect to the metabolic activity of HDFs seeded in the absence of particles, which was used as a control. The presence of BaSO_4_ and PD-BaSO_4_ particles slightly reduced the metabolic activity of HDFs with respect to the control. However, in all the cases, the metabolic activity was higher than 80%, demonstrating that HDFs were able to maintain a normal metabolic activity in the presence of particles (see also Appendix A). At day 1, the metabolic activity of HDFs seeded with 10, 50, 100 and 500 µg/mL of BaSO_4_ particles was, 95, 94, 90 and 90%, respectively, relative to that of the control. In the case of HDFs seeded with 10, 50, 100 and 500 µg/mL of PD-BaSO_4_ particles, the metabolic activity was 89, 86, 89 and 83% that of the control, respectively. At day 3, the metabolic activity of those cells seeded with 10 or 50 µg/mL of BaSO_4_ particles, as well as 10 µg/mL of PD-BaSO_4_ particles was not significantly different from the metabolic activity of the control. At this day, the metabolic activity in all the cases was higher than 90%, suggesting a negligible effect of the particles in the metabolic activity of HDFs. Additionally, cells observed under an inverted microscope showed normal morphology and the BaSO_4_ and PD-BaSO_4_ particles seemed to be internalized by cells and located around the nuclei (Appendix A).

Figure 4 shows metabolic activity of cells seeded in PLA, PLA/BaSO_4_, PLA/PD-BaSO_4_ with 10 wt.% of filler composites and control tissue culture plastic (TCP). The metabolic activity was normalized at each time-point with the metabolic activity of HDFs seeded onto the TCP, which was used as a control. Except for cells seeded on PLA after 3 days of culture (see Appendix A), no significant differences were observed in the metabolic activity of HDFs with respect to the control, indicating normal metabolic activity of cells seeded on the composites developed in this work. 

The proliferation of HDFs on PLA, PLA/BaSO_4_ and PLA/PD-BaSO_4_ composites was evaluated via DNA quantification. As can be seen in Figure 5, DNA content increased over culture time in all experimental and control conditions. Accordingly, significant differences were observed between DNA content observed at day 7 and that observed at day 1 for all the samples studied. For example, the calculated proliferation rates between day 1 and day 7 were 1.6, 3.2 and 2.4 for PLA, PLA/BaSO_4_ and PLA/PD-BaSO_4_, respectively. The metabolic activity and proliferation results demonstrate that the materials employed in this work are not cytotoxic and can provide a cytocompatibility substrate for cells to attach and proliferate. A higher proliferation of HDFs was observed in PLA/BaSO_4_ or PLA/PD-BaSO_4_ composites with respect to pristine PLA samples. Is hypothesized that this higher proliferation rate may be associated to surface characteristic of the samples, such as roughness or hydrophilicity [35,36]

Concurrently, PD coating of BaSO_4_ particles introduced binding sites for biologically active molecules such as proteins or drugs, increasing the use of this composite in second-generation devices for biomedical applications.

### 2.3. Adsorption/Release Test in 3D Printed Scaffolds

Following confirmation of cytocompatibility, the potential for antibiotic delivery with 3D printed scaffolds of PLA/PD-BaSO_4_ was evaluated in vitro, with the aim of preventing an infection due to the insertion surgery (open wound) and the consequent rejection of the device [37]. To this end, levofloxacin was incorporated into the material via PD-BaSO_4_ particle functionalization as a local drug delivery system for avoiding the oral administration common in this kind of surgeries. Levofloxacin is used for fighting and preventing osteomyelitis, since it is a fluoroquinolone with anti-staphylococcal activity in osteoarticular tissues [38].

Here, in order to compare the levofloxacin loading efficiency of the developed composites a polydopamine coated neat PLA sample (termed PD-PLA) is introduced. The release was performed at pH 5 to simulate the state of infection and at body temperature of 37 °C.

Figure 6 shows a non-detectable release of PLA, this is because the scaffolds have been previously washed, therefore there was a very low level of drug in PLA scaffolds, reflecting also that PLA does not adsorb the antibiotic and consequently does not release it either. Furthermore, in the case of PD-PLA a burst release was observed during the first 24 h where almost 80% of levofloxacin eluted. The remaining amount of drug (20%) was eluted in 4 days. This type of release profile is adequate in the context of an infection, since at the beginning a high rate of drug release is desirable followed by a slower drug release. When the PLA/PD-BaSO_4_ was analyzed, a more moderate burst release was observed where almost 60% of levofloxacin eluted and later, during the next 3 days the 30% of drug was eluted while the last two days shows a very slow release with an 8% release (see Appendix A). In the end the PLA only releases 0.43 µg/mL due to the drug that is trapped in the holes. While PD-PLA was the one with the highest amount of drug released, 3.12 µg/mL, the PLA/PD-BaSO_4_ composite drug released a 2.4 µg/mL. 

These release results in global show that PD-PLA and PLA/PD-BaSO_4_ display potential antimicrobial properties. Comparing these two materials, the release profile of PLA/PD-BaSO_4_ composites was more interesting as it generated a second elution stage with greater release, 40% versus 20% in PD-PLA. It is remarked finally that these scaffolds were immersed in trizma buffer (pH 10) prior to analysis so the results of Figure 6 correspond to bound levofloxacin after adsorption.

### 2.4. Antimicrobial Activity of 3D PLA/PD-BaSO_4_ Scaffolds with Levofloxacin

To analyze the antibiotic efficacy of the 3D printed scaffolds the Agar Disk Diffusion tests against *Staphylococcus aureus (S. aureus)* were carried out. *S. aureus* was chosen because in bone infections is one of the most important pathogens due to its ability to adhere and form biofilms when in contact with tissue [38]. Figure 7 shows the Agar disk diffusion test corresponding to 3D printed scaffolds of PLA/PD-BaSO_4_ with levofloxacin, PLA/PD-BaSO_4_ scaffold as a negative control without levofloxacin and the disk of levofloxacin (5 µg) as a positive control. Cicuéndez et al. calculated that the minimum inhibitory concentration (MIC) of levofloxacin has a value of 0.06 μg/mL [39]. As observed in the release assay (Figure 6), these scaffolds release larger amounts of the drug.

Analyzing the Agar Disk Diffusion tests is observed that PLA/PD-BaSO_4_ scaffolds with levofloxacin effectively inhibit bacterial growth, being the inhibition zone diameters of 38 ± 4 mm (Figure 7a), while the positive control (5 µg of Levofloxacin) exhibited a diameter average 28 ± 0 mm (Figure 7c). No inhibition zone could be observed for the sample without levofloxacin, PLA/PD-BaSO_4_ (negative control, Figure 7b). It should be noted here that the scaffolds were washed after the drug adsorption, therefore the effective inhibition of bacterial growth is attributed to the amount of levofloxacin bound to polydopamine coating of the particles and not to an unspecified content of drug that could remain in the scaffold holds. This would still lead to a higher inhibition zone. 

From Agar diffusion tests can be concluded that the polydopamine coating method of BaSO_4_ particles used for drug tethering within the PLA composites is effective and facilitates the release of the drug inhibiting the *S. aureus* growth. The described method is simple and allows the addition of the antibiotic after a 3D printing process of scaffolds. This fact has two advantages since the drug is not processed with the scaffold avoiding either the contact with organic solvents nor high processing temperatures that could degrade the molecules, and, on the other hand, the adsorption of the medicament by a scaffold or implant can be done when clinically necessary, in situ.

## 3. Materials and Methods

### 3.1. Materials

Poly (D-lactide) homopolymer (100,000 gmol^−1^) (PLA) was supplied by Purac-Corbion (Barcelona, Spain). Dopamine chloride, Barium sulfate (BaSO_4_), phosphate buffer saline (PBS) and levofloxacin were purchased from Sigma-Aldrich (Madrid, Spain). Dulbecco’s modified Eagle´s medium (DMEM), fetal bovine serum (FBS), Hank´s balanced salt solution (HBSS) and penicillin-streptomycin (PS) solution Human dermal fibroblasts (HDFs) were purchased from Sigma-Aldrich (Arklow, Ireland). Quant-ITTM PicoGreen^®^ dsDNA kit was from Invitrogen (Dublin, Ireland) and AlamarBlue^®^ is from ThermoFisher Scientific (Dublin, Ireland).

### 3.2. Coating with Polydopamine and Blending

Coating of particles covered with polydopamine (PD) to obtain PD-BaSO_4_ were obtained as in our previous article using basic pH 8.5 for 24 h followed by filtration and drying in a vacuum oven overnight [17]. The same coating method was used for the PD-PLA scaffolds. The size of the particles was measured by HORIBA Laser Scattering Particle Size Distribution Analyzer LA-350 (Horiba, Kyoto, Japan).

Polylactide samples were filled with 0.5, 1, 2, 5 and 10 wt.% of PD-BaSO_4_. PD-BaSO_4_ particles and neat polylactide (PLA) were melt mixed in a DSM Xplore micro-compounder (Xplore Instruments, Sittard, The Netherlands) at 200 °C and 150 rpm during 2 min and then conformed by injection molding. The mold was pre-heated at 45 °C and the injection temperature was 200 °C. Gel Permeation Chromatography (GPC) tests were carried out before and after blending process to check that the matrix is not degraded (see Appendix A). The weight average molecular mass (Mw) and dispersity (D) of PLA pellets before processing were respectively Mw = 183758 gmol^−1^ and D = 1.8 whereas, after processing in a PLA/BaSO_4_ composite, the values obtained for PLA were Mw = 174980 gmol^−1^ and I = 1.8.

Scanning Electron Microscopy (SEM) and Transmission Electron Microscopy (TEM) were used to observe both the degree of dispersion of the particles inside of the polymer matrix and the break surfaces.

### 3.3. 3D Printing of Radiopaque Scaffolds

Scaffold fabrication was carried out in a 3D-Bioplotter from Envision TEC. Scaffolding 3D model was designed with Computer Aided Drawing (CAD, Autodesk, San Rafael, CA, USA). The CAD model was uploaded to the Bioplotter software (version 3.0.713.1406, Envision TEC, Gladbeck, Germany), which enables slicing of the model, before 3D-printing.

Two types of scaffolds were designed: one for mechanical properties and the other for the antibiotic release tests. Cylindrical geometries having 10 mm diameter and 10 mm height were used for measuring mechanical properties in compression mode, whereas square geometries (10 × 10 × 2 mm) (length × width × height) were used for adsorption/release and Agar Diffusion tests. In both cases the pore size of the scaffolds was 500 µm and it was used a plastic conic needle of 0.25 mm inner diameter for the printing layer dimension (Appendix A). The resulting file was uploaded to the software Visual Machines (version 2.8.126) that allows the user to input the various parameters that control the Bioprinter (Envision TEC).

Due to the specific dimensions of the feeding cartridge and the degradable nature of the polymer, the scaffolds are printed by solution. The materials (PLA and PLA/PD-BaSO_4_) were dissolved with chloroform during 48 h, printed at 20 °C. The printing conditions are shown in Table 2. Reproducibility of the PLA and PLA/PD-BaSO_4_ support frames was ensured by using the same CAD model for each frame, and by the high XYZ axis resolution of the Bioplotter (0.001 mm).

### 3.4. Mechanical Properties

Dumbbell-shaped samples for tensile tests were punched out from sheets following ISO 527-2 (ISO 527-2/5A/5). In the case of 3D-printed scaffolds, the mechanical properties under compression were obtained following ISO 604. In both testing modes the tests were performed with an Instron 5565 testing machine at 23 °C and 50% of relative humidity (RH). 

### 3.5. Adsorption/Release Test

Before starting the release test, the adsorption of levofloxacin in the scaffolds has been carried out. For that, the scaffolds were submerged in a buffer at pH 10 with levofloxacin (2 mg/mL) [40] for 24 h for facilitate the adsorption. Then, the scaffolds were washed with buffer at pH 10 and dried. The washing was done to eliminate the drug from the holes of the scaffolds, and in this way obtain the amount of drug that is absorbed by the material itself.

Levofloxacin release was evaluated using a UV-vis spectrophotometer (BMG Labtech FLUOstar Omega, Ortenberg, Germany). The band in λ = 287 nm for levofloxacin was employed to build a standard curve with known concentrations of the drug in PBS and the measurements were carried out at concentrations lower than 25 ppm. Release of levofloxacin from the scaffolds was tested in 3 mL of PBS at 37 °C, under mild agitation (220 rpm) at pH 5. Samples were taken at the following times: 30 min, 1 h, 2 h, 4 h, 6 h, 8 h, 12 h, 24 h, 48 h 72 h, 96 h and 144 h. In all materials, three independent tests were carried out.

### 3.6. Agar Disk Diffusion

The diffusion disk agar tests for *Staphylococcus aureus* inoculum were performed under the Clinical and Laboratory Standards (CLSI) [40]. Briefly, few colonies of *S. aureus* were resuspended in Mueller Hinton Broth (Biokar Diagnostics, Pantin, France) and further diluted in order to achieve 0.5 McFarland units (1 × 10^8^ CFU mL^−1^) at 600 nm of wavelength, using a spectrophotometer (U-2000, Hitachi, Tokyo, Japan). The inoculum was swabbed on Mueller Hinton Agar (Biokar Diagnostics) plates and the scaffolds (one without drug as a negative control) were tested, as well as 5 µg of levofloxacin disk (positive control). Petri dishes were further incubated (Ultima, Revco, Thermo Scientific) at 37 °C for 24 h. Assays were performed in three independent experiments

### 3.7. Cell Culture

#### 3.7.1. Human Dermal Fibroblasts

HDFs were grown in T75 flasks using DMEM with 10% FBS and 1% PS. The cells were incubated at 37 °C in an atmosphere of 5% CO_2_. The culture medium was changed every 3 days. The cells were then harvested and sub-cultured when >90% confluence was observed.

#### 3.7.2. Cell Seeding

To study the metabolic activity of HDFs seeded in the presence of BaSO_4_ or PD-BaSO_4_, HDFs were seeded at a density of 25,000 cells/cm^2^ in a 96 well tissue culture plate and incubated in 0.5 mL of DMEM with 10% FBS and 1% PS (37 °C, 5% CO_2_). After 1 day in culture, the media was replaced by complete media containing 0, 10, 50, 100 or 500 µg/mL of BaSO_4_ or PD-BaSO_4_ particles that were previously autoclaved.

HDFs were also seeded on sterilized PLA, PLA/BaSO_4_ and PLA/PD-BaSO_4_ samples (7 mm in diameter) at a density of 25,000 cells/cm^2^ in a 48 well tissue culture plate and incubated in DMEM with 10% FBS and 1% PS. First, HDFs were suspended in 40 µL of culture medium, seeded onto each sample and incubated for 2 h to allow cell attachment (37 °C, 5% CO_2_, 95% relative humidity). When cells were attached, and additional 0.5 mL of culture medium was added into each well. The culture medium was replaced at day 3 after seeding.

#### 3.7.3. Cell Viability Studies

AlamarBlue^®^ assay was performed to quantify the metabolic activity of HDFs in the presence of BaSO_4_ or PD-BaSO_4_ particles or seeded on PLA, PLA/BaSO_4_ and PLA/PD-BaSO_4_. At the selected time points (1, 2 and 3 days for HDFs in the presence of BaSO_4_ or PD-BaSO_4_ particles and 1, 3 and 7 days for HDFs seeded on PLA, PLA/BaSO_4_ and PLA/PD-BaSO_4_), the cells were washed with HBSS and subsequently incubated (6 h, 37 °C, sheltered from light) in 0.5 mL of fresh culture media with AlamarBlue^®^ (10% *v*/*v*). Then, 100 µL of assay media was transferred to a 96 well plate, the absorbance at 550 and 595 nm was read on a microplate reader (Varioskan Flash, Thermo Fisher Scientific) and the percentage reduction of the dye was calculated.

To quantify the DNA amount of cells seeded on PLA, PLA/BaSO_4_ and PLA/PD-BaSO_4_ a PicoGreen^®^ assay was then performed on the same samples used for AlamarBlue^®^. Cells were repeatedly frozen at −80 °C and thawed to lyse the cells and release the entire DNA content. Finally, fluorescence was measured at 480 nm.

#### 3.7.4. Statistics

Statistical differences were analyzed using one-way analysis of variance (ANOVA) and *p*-values of <0.05 were considered significant. Experiments were performed in triplicate and each assay was repeated three times.

## 4. Conclusions

In this work the mechanical brittleness associated to polylactide homopolymers is overcome via the incorporation of PD-BaSO_4_ particles into a PLA matrix. The composite materials showed enhanced stiffness, strength, ductility and toughness. This is indeed relevant since in classical polymer/inorganic composites increases in stiffness and strength usually lead to dramatic decreases of ductility and toughness.

The particulate composites studied in this work are proved to be a valid substrate for cells to attach and proliferate. In addition, coating of barium sulfate particles with polydopamine provides functional groups that can act as anchorage points for incorporation of molecules with biological activity i.e., levofloxacin which is an antibacterial drug. The benefits of PD-BaSO_4_ prove the potential of use of these PLA composites in bone reconstruction applications.

## Figures and Tables

**Figure 1 ijms-21-05480-f001:**
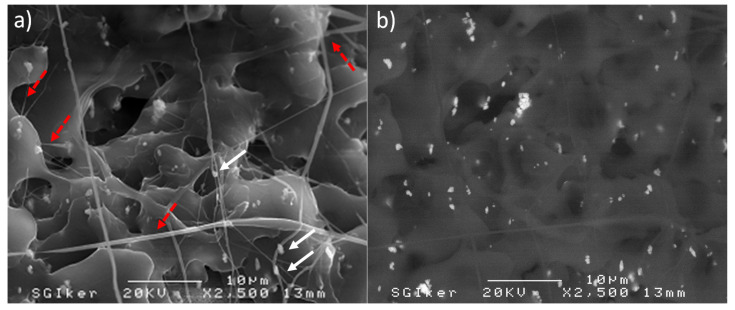
SEM image of PLA/PD-BaSO_4_ 2 wt.% of filler composite obtained with dispersion of (**a**) secondary electrons and (**b**) backscattered electrons.

**Figure 2 ijms-21-05480-f002:**
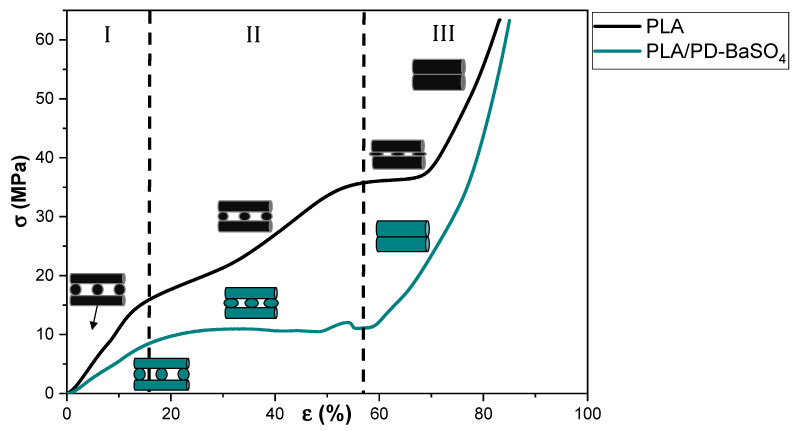
Compression stress (σ)–strain (ε) curves of scaffolds for neat polylactide (PLA) and scaffolds for composite of polylactide and coated with polydopamine barium sulfate particles (PLA/PD-BaSO_4_) 10 wt.%.

**Figure 3 ijms-21-05480-f003:**
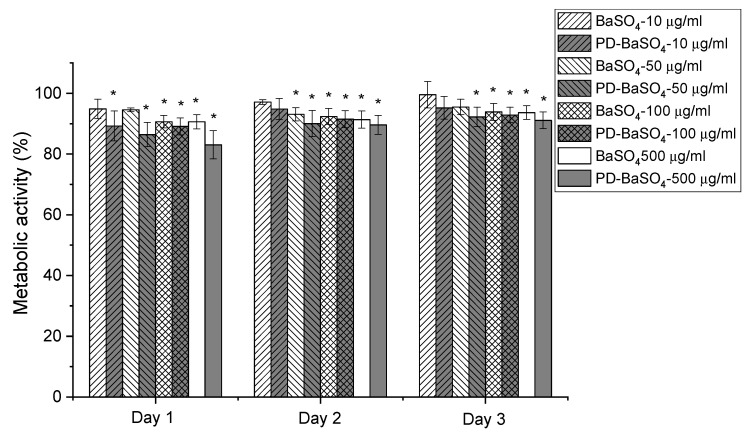
Metabolic activity of Human dermal fibroblasts (HDFs) seeded in the presence of 10, 50, 100 or 500 µg/mL of barium sulfate particles (BaSO_4_) or barium sulfate particles coated with polydopamine (PD-BaSO_4_). Asterisks indicate significant differences (*p* < 0.05) with respect to the control.

**Figure 4 ijms-21-05480-f004:**
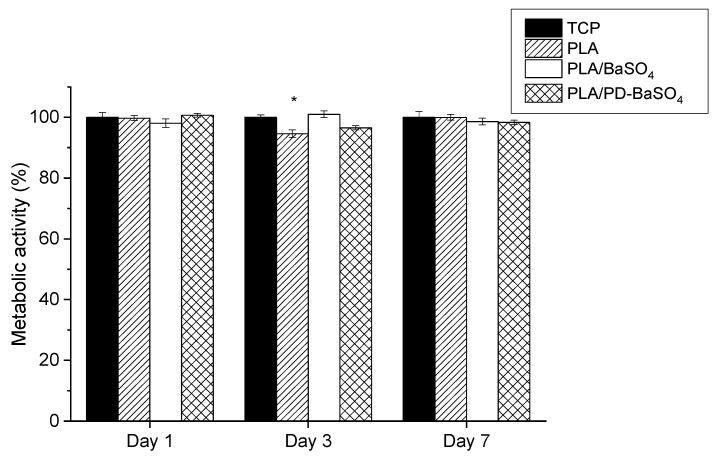
Metabolic activity of Human dermal fibroblasts HDFs seeded on polylactide (PLA), composite of polylactide and barium sulfate particles (PLA/BaSO_4_) and composite of polylactide and coated with polydopamine barium sulfate particles (PLA/PD-BaSO_4_) with respect to the control at day 1, 3 and 7. Asterisks indicate significant differences (*p* < 0.05) with respect to the cells seeded on tissue culture plastic (TCP).

**Figure 5 ijms-21-05480-f005:**
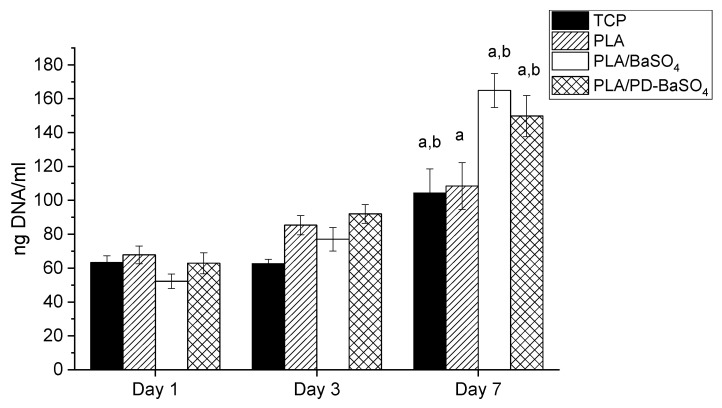
Proliferation of Human dermal fibroblasts HDFs seeded polylactide (PLA), composite of polylactide and barium sulfate particles (PLA/BaSO_4_) and composite of polylactide and coated with polydopamine barium sulfate particles (PLA/PD-BaSO_4_) a and b indicate significant differences (*p* < 0.05) with respect to day 1 and day 3, respectively.

**Figure 6 ijms-21-05480-f006:**
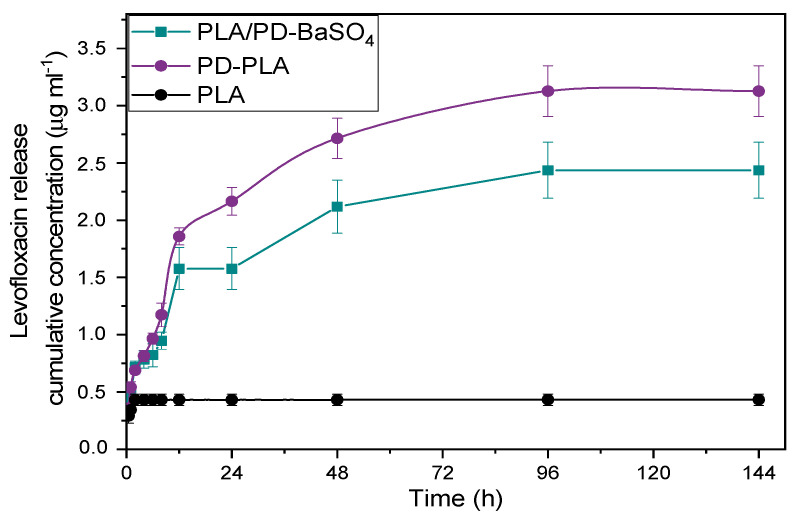
Release profiles over time of levofloxacin in neat PLA, polydopamine coated neat PLA (PD-PLA) and PLA/PD-BaSO_4_.

**Figure 7 ijms-21-05480-f007:**
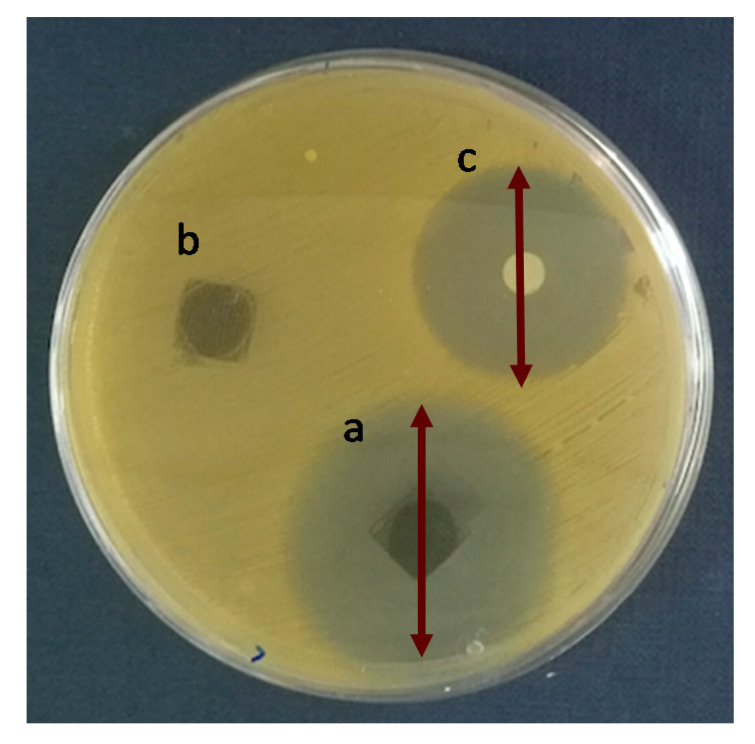
Agar disk diffusion tests: (**a**) PLA/PD-BaSO_4_ with levofloxacin, (**b**) PLA/PD-BaSO_4_ negative control (no levofloxacin) and (**c**) levofloxacin disk (5 µg) as a positive control.

**Table 1 ijms-21-05480-t001:** Mechanical properties of neat polylactide (PLA) and PLA/polydopamine-barium sulfate (PD-BaSO_4_) composites with respect to wt.% of PD-BaSO_4_. σ_y_: yield stress, E: Young’s modulus, σ_b_: stress at break, ε_b_ elongation at break and TT: tensile toughness.

PD-BaSO_4_ (wt.%)	E (MPa)	σ_y_ (MPa)	σ_r_ (MPa)	ε_b_ (%)	TT (J/m^3^)
0	1243 ± 96	-	67.1 ± 0.5	9.2 ± 0.5	3.8 ± 0.6
0.5	1309 ± 51	78.7 ± 0.4	56.3 ± 5.4	132.2 ± 13.1	67.5 ± 6.1
1	1396 ± 109	77.7 ± 0.8	56.0 ± 4.8	146.3 ± 9.8	73.6 ± 4.6
2	1417 ± 86	78.4 ± 1.7	62.5 ± 5.2	182.2 ± 5.9	95.1 ± 1.7
5	1415 ± 49	75.7 ± 0.6	57.6± 5.8	171.9 ± 2.9	85.9 ± 0.8
10	1350 ± 124	74.1 ± 0.8	55.3 ± 4.3	154.6 ± 8.6	76.2 ± 2.8

**Table 2 ijms-21-05480-t002:** Printing conditions for neat PLA and composite PLA/PD-BaSO_4_.

Material	Temperature (°C)	Pressure (Bar)	Speed(mm/s)	Post-Flow(s)	Pre-Flow(s)
PLA	25	5.0	3.5	0.11	0.04
PLA/PD-BaSO_4_	25	4.4	4.1	0.11	0.01

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
