# Peer review of "Benefits of Polydopamine as Particle/Matrix Interface in Polylactide/PD-BaSO4 Scaffolds"

_ijms, 2020, doi:10.3390/ijms21155480_

Round 1
Reviewer 1 Report
The present study described the physicochemical characterization and in vitro evaluation of a polylactide scaffold with barium sulfate particles functionalized with polydopamine. In general, the physicochemical and in vitro tests showed that the biomaterial seems to be a promising material. However, the manuscript contains some spelling and grammatical errors. Moreover, the clinical application of the biomaterial proposed need to be clarified for a better evaluation of the methodology used in the study. I suggest a broad revision of the manuscript before it is accepted for publication.
- All abbreviations used in the text need to be described first. Please, revised all text.
1) Introduction:
1A) The introduction section needs improvements:
- Line 31 – 31: Confused phrase;
- Line 37 – 39: It is suggested that the aim of the study be inserted only at the introduction end;
- Line 40 – 47: Confused paragraph;
- Line 52- 56: I suggest removing this text once the citations are outside the study context.
- Line 57 – 69: Some abbreviations are not described in the text (BMP-2, IGF-1, MWCNT, PU, PD, DP-MWCNT)
- Line 67 – 69: Confused phrase;
1B) What is the clinical application of the proposed material? This information must be clear to the reader.
2) Results and discussion
2A) The Results and discussion section needs improvements:
- Table 1 must first be described and discussed;
- Line 81 – 84: Confused phrase;
- Line 90 – 91: Confused phrase;
- Line 94 – 97: Confused phrase;
- Line 113 – Replace "Figure 1a" for "Figure 1"
- Line 118: Add a point before "Therefore";
- Line 145 – 146: The word "also" is correct? Please revised this phrase;
- Line 235 – 238: Repeated information. I suggest that this section be removed and replaced by the line 243-247;
2B) Why the Human dermal fibroblasts (HDFs) was used in this study? This information needs to be clear for the reader in the discussion section;
2C) Line 201 – 202: If the material is biocompatible, I do not believe it is necessary to incorporate an antibiotic to avoid rejection of the material. The incorporation of the antibiotic needs to be better justified. Please explain and revised this phrase;
2D) What was the final concentration released of levofloxacin after 144 hours for each scaffold tested? Please add this information.
2E) The 3D printing way of the scaffolds was not discussed. Please include more information about that.
2F) More published studies need to be incorporated into the discussion to compare the results obtained.
3) Materials and Methods
3A) The Material and Methods section needs improvements:
- Line 258: Correct the section number;
- Line 261: Change "dopamine chloride" for "Dopamine chloride";
- Line 285: Remove ")"
- Line 292 - 295: The printing conditions for PLA/BaSO4 scaffold was not described in Table 2.
- Line 302 - 307
- Line 320: add the term "disk" in "5 mg of levofloxacin (positive control)".
3B) The sequence of the "Materials and methods section" need to be the same as the "Results and discussion" section. Therefore, the "Cell Viability Studies" need to be described before the "Adsorption/Release test".
3C) The scaffolds used in each experimental text need to be added.
3D) The methodology for "polydopamine coated neat PLA sample" production used in Adsorption/Release test (line 206) was not described in "Materials and methods" section.
3E) Why the PLA/PD sample was not tested in cell viability studies?
3F) Why the concentration of levofloxacin used in the study was 2 mg/mL?
3G) The ANOVA one-way was used in each experiment? The statistical analysis must be performed on all the tests described in the study.
4) Conclusion
- Line 367 – 379: This information can be removed.
4A) Which scaffolds performed best in all tests? Please add this information in the conclusion section.
Author Response
Response to Reviewer 1 Comments
First of all, thanks to the reviewers for helping us improve our paper with their constructive comments. We have carefully considered all of them and the changes have been performed accordingly.
The comments of the reviewers are answered individually and specifically (in bold)
The present study described the physicochemical characterization and in vitro evaluation of a polylactide scaffold with barium sulfate particles functionalized with polydopamine. In general, the physicochemical and in vitro tests showed that the biomaterial seems to be a promising material. However, the manuscript contains some spelling and grammatical errors. Moreover, the clinical application of the biomaterial proposed need to be clarified for a better evaluation of the methodology used in the study. I suggest a broad revision of the manuscript before it is accepted for publication.
- All abbreviations used in the text need to be described first. Please, revised all text.
Previous to the abbreviations the full name has been written in all cases along the text.
- Introduction:
1A) The introduction section needs improvements:
- Line 31 – 31: Confused phrase;
The mentioned phrase has been substituted for the next one.
- Line 37 – 39: It is suggested that the aim of the study be inserted only at the introduction end;
The aim of the study has been placed at the introduction end.
- Line 40 – 47: Confused paragraph;
We have rewritten the paragraph to make it clear:
“Inorganic particulate reinforcements can enhance the mechanical properties of polymers and confer additional filler-specific properties to the matrix [11]. It is also well known that the enhancement of strength related mechanical properties (elastic modulus, resistance to yield and ultimate stress) is much more noticeable with continuous fiber composites[12], however continuous fiber formulations do not lend themselves well to free-form manufacturing techniques such as extrusion, injection molding or, as is the case, 3D printing [13–15]. Particularly, 3D printing is progressing very rapidly as an advanced manufacturing technique also for particulate composites. In this domain, there are also considerable achievements in the development of drug delivery systems. Therefore, in this work we selected a 3D printable system based on a BaSO4 particulate and PLA composites with an antimicrobial drug incorporation for the potential use in fixation devices for bone tissue regeneration”
- Line 52- 56: I suggest removing this text once the citations are outside the study context.
We agree with this suggestion for these works are outside the study context. The paragraph has been removed.
- Line 57 – 69: Some abbreviations are not described in the text (BMP-2, IGF-1, MWCNT, PU, PD, DP-MWCNT)
Descriptions of abbreviations have been incorporated into the text in their corresponding places:
“[…] bone morphogenic protein 2 (BMP-2), insulin-like growth factor 1 (IGF-1), multi-walled carbon nanotubes (MWCNT), polyurethanes (PU) […] .”
- Line 67 – 69: Confused phrase;
The sentence has been modified to: “Few works have reported changes in mechanical properties of polymer composites due to polydopamine coating of the inorganic phase, one of them is that on polylactide reinforced by polydopamine functionalized halloysite nanotubes [29].”
1B) What is the clinical application of the proposed material? This information must be clear to the reader.
The specific clinical application (bone fixation devices) for this material has been added in abstract (line 24), introduction, and conclusion sections.
2) Results and discussion
2A) The Results and discussion section needs improvements:
- Table 1 must first be described and discussed;
All the paragraphs related to the results and their discussion of table 1 have been moved ahead of the table 1. The results presentation and discussion have been improved as needed.
- Line 81 – 84: Confused phrase;
A new phrase is written in substitution
“As identified in our previous works with PLA/BaSO4 composites [3,5], the plastic deformation in composites occurs because the rigid particles act as stress concentrators and, after particles debonding from the matrix and being of a 0.7-1.9 µm size [27], activate, at a point of applied stress, crazing and shear yielding deformation mechanisms. Since polydopamine covered particles (PD-BaSO4) present 1.25 µm size and interactions with the matrix through the polydopamine interface, in these composites, increases in all mechanical properties (elastic modulus, strength, ductility and toughness) are achieved [28].”
- Line 90 – 91: Confused phrase; and - Line 94 – 97: Confused phrase;
The paragraph has been rewritten as follows to avoid confusions.
“Table 1 shows the tensile mechanical properties of PLA/PD-BaSO4 composites. As can be observed, up to a 2 wt. % BaSO4 content in composites, Young’s modulus, ductility and toughness improvements can be observed. Further, if these results are to be compared with those of PLA/BaSO4, the non PD functionalized composite counterparts, enhancements in stiffness and strength together with ductility and toughness can be noticed in the former. Please also note that a huge > 2300 % increase in elongation at break are determined in these novel composite formulations in regard to neat PLA, which brings about a dramatic improvement in toughness. This is attributable to specific interactions stablished between the ester groups of polylactide and the alcohol groups of polydopamine coating of the particles surface [18,29] bringing about a stronger fiber/matrix interface. It is also noticeable that a 30% increase in the elongation at break is obtained in PD coated BaSO4 PLA composites in regard to the composites without polydopamine coating [3]. Finally, beyond the 2 wt. % particle composition, a slight general decrease in mechanical properties are noticeable, suggesting the existence of particle aggregates and bundles beyond this point.”
- Line 113 – Replace "Figure 1a" for "Figure 1"
Done
- Line 118: Add a point before "Therefore";
Done
- Line 145 – 146: The word "also" is correct? Please revised this phrase;
It has been removed ‘also’ from the text.
- Line 235 – 238: Repeated information. I suggest that this section be removed and replaced by the line 243-247;
Done
2B) Why the Human dermal fibroblasts (HDFs) was used in this study? This information needs to be clear for the reader in the discussion section;
Fibroblasts were employed as a model cell type for the preliminary in vitro studies considering that they are the most abundant cell type of connective tissues. Besides, fibroblast have been thoroughly reported in bibliography as a well-established tool to evaluate the cytotoxicity of biomaterials. More precisely, in this study fibroblasts from human origin were considered, thus providing a highly relevant cell source to study the interaction between cells and biomaterials.
Ideally the work would have been better if performed with bone cells, either osteoclasts, osteoblasts or even bone-marrow stem cells. Yet the work with fibroblasts is a significant contribution at the point of the research of this new system.
A new sentence has been stated for a clearer understanding is incorporated at the beginning of the 2.2. Biocompatibility assessment section in line 153: ‘In vitro compatibility studies were performed to determine the possible toxicity of BaSO4 and PD-BaSO4 particles and their resulting PLA composites using Human dermal fibroblasts (HDFs) as a basic toxicity test”
2C) Line 201 – 202: If the material is biocompatible, I do not believe it is necessary to incorporate an antibiotic to avoid rejection of the material. The incorporation of the antibiotic needs to be better justified. Please explain and revised this phrase;
To better justify the necessity of incorporation the antibiotic the following paragraph has been rewritten in section 2.3. Adsorption/Release test in 3D printed scaffolds.
“Following confirmation of cytocompatibility, the potential for antibiotic delivery with 3D printed scaffolds of PLA/PD-BaSO4 was evaluated in vitro, with the aim of preventing an infection due to the insertion surgery (open wound) and the consequent rejection of the device [36]. To this end, levofloxacin was incorporated into the material via PD-BaSO4 particle functionalization as a local drug delivery system for avoiding the oral administration common in this kind of surgeries. Levofloxacin is used for fighting and preventing osteomyelitis, since it is a fluoroquinolone with anti-staphylococcal activity in osteoarticular tissues [37].”
2D) What was the final concentration released of levofloxacin after 144 hours for each scaffold tested? Please add this information
PLA 0.43 µg/ml PD-PLA 3.12 µg/ml and PLA/PD-BaSO4 2.4 µg/ml
We add this information in the line 223:
“In the end the PLA only releases 0.43 µg/ml due to the drug that is trapped in the holes. While PD-PLA is the one with the highest amount of drug released, 3.12 µg/ml, the PLA/PD-BaSO4 composite drug release was a 2.4 µg/ml.”
2E) The 3D printing way of the scaffolds was not discussed. Please include more information about that.
The 3D printing way is presented in section 3.3. 3D printing of radiopaque scaffolds (now in lines 285-302). A sentence has been added in lines 297-298 in the revised version. “Due to the specific dimensions of the feeding cartridge and the degradable nature of the polymer, the scaffolds are printed by solution”
2F) More published studies need to be incorporated into the discussion to compare the results obtained.
Done.
“after particles debonding from the matrix and being of a 0.7-1.9 µm size [27], activate, at a point of applied stress, crazing and shear yielding deformation mechanisms”
“with the aim to preventing an infection due to the insertion surgery (open wound) and the consequent rejection of the device [36] “
“For that, the scaffolds were submerged in a buffer at pH 10 with levofloxacin (2 mg/mL)[39] for 24 h for facilitate the adsorption”
3) Materials and Methods
3A) The Material and Methods section needs improvements:
- Line 258: Correct the section number;
Done
- Line 261: Change "dopamine chloride" for "Dopamine chloride";
Done
- Line 285: Remove ")"
Done
- Line 292 - 295: The printing conditions for PLA/BaSO4 scaffold was not described in Table
We have corrected the mistake in the text writing the correct system PLA/PD-BaSO4, instead of PLA/BaSO4.
- Line 302 – 307
There is no appointment in these lines, we have carefully check it and have not found anything to change.
- Line 320: add the term "disk" in "5 mg of levofloxacin (positive control)".
Done
3B) The sequence of the "Materials and methods section" need to be the same as the "Results and discussion" section. Therefore, the "Cell Viability Studies" need to be described before the "Adsorption/Release test".
Done
3C) The scaffolds used in each experimental text need to be added.
In the revised version, the geometry of the two types of scaffolds used is described.
3D) The methodology for "polydopamine coated neat PLA sample" production used in Adsorption/Release test (line 206) was not described in "Materials and methods" section.
Done, we have included it in the methods section in line 273: ‘The same coating method was used for the PD-PLA scaffolds.’ According to this, the title of the section has been changed to: 3.2. Coating with polydopamine and blending
3E) Why the PLA/PD sample was not tested in cell viability studies?
The biocompatibility of polylactide and polydopamine, by their own, is well known and reported. The novelty of the work focuses in the polydopamine coating of BaSO4 that leads to beneficial interphase interactions and improvement in mechanical properties providing also functional groups that can act as anchorage points for incorporation of other molecules with biological activity.
3F) Why the concentration of levofloxacin used in the study was 2 mg/mL?
We chose a high drug concentration for assuring the maximum adhesion of the drug to the scaffolds.[39]
3G) The ANOVA one-way was used in each experiment? The statistical analysis must be performed on all the tests described in the study.
The ANOVA one-way has been performed in cell viability tests. In mechanical tests, the average values of 5 specimens are displayed, and for release and microbiological tests the averages are taken as the mean value of 3 samples.
4) Conclusion
- Line 367 – 379: This information can be removed.
Rewritten.
4A) Which scaffolds performed best in all tests? Please add this information in the conclusion section.
The second paragraph of the conclusion has been rewritten and now states as follows:
“The particulate composites studied in this work are proved to be a valid substrate for cells to attach and proliferate. Besides, coating of barium sulfate particles with polydopamine provides functional groups that can act as anchorage points for incorporation of molecules with biological activity i.e. levofloxacin which is an antibacterial drug. The benefits of PD-BaSO4 prove the potential of use of these PLA composites in bone reconstruction applications.”
Reviewer 2 Report
The authors describe a composite system based on a PLA matrix with incorporated polydopamin-coated BaSO4 particles. By combining these structural features, several points are addressed that are of relevance for application of polymers as bone implants. The benefits include: radiopacity, increased toughness while increasing the elongation at break, and therefore less brittle materials, and the ability of increasing drug loading and controlling the release.
While the article is allover well written, I have two major concerns, a minor concern, and several small remarks.
- My main concern 1 is the choice of PLA as matrix material. While there are probably hundreds of articles regarding the use of PLA as implants and drug releasing devices, especially for bone applications PLA and derivatives are questionable as major component. Since a long time it is known that while such PLA (and related polyesters) implant materials may help in the initial phase of bone regeneration, the subsequent acidification of the bone tissue because of material degradation may lead to problems in the clinic. See for example J. Biomed. Mater. Res. 1997, 38: 105-114 or also Biomaterials 2004, 25, 5735. This is why generally these days not major bone replacements are made from such materials, but may be small additions such as screws or drug releasing particles, or extra measures are undertaken to make sure that the acidification is counteracted by coatings, additions etc. So, ideally for this study, a different polymer matrix should have been chosen. As this is a bit late now, the authors should address this point with relevant literature, perhaps defining the exact application of their potential implant material. Also, it could be discussed whether the BaSO4 or PD may positively contribute, though their influence will be small due to the low content. In the end, the potential impact of the work is hampered by the material selection.
- Major concern 2: The composites are formed at 200 °C. If you have ever performed a DSC and/or TGA of PLA, you will have noticed that at this temperature the degradation of PLA already starts. It is necessary to provide data of the PLA before/after composite formation (NMR, GPC) to show that the PLA did not degrade and that you actually have the molar mass that you claim.
- Minor concern: If you already have specified that your materials shall be implanted in bone, why have you studied the influence on fibroblasts, rather than performing experiments with osteoblasts (and perhaps also osteoclasts)? True, very often hibroblasts are used for basic toxicity tests, but such other work would increase the validity of the work a lot.
Small remarks
- Line 4-42: The sentence should be split in two after ref. [3,5]
- Line 82. Again, start a new sentence after …system [3,5]. The yield…
- Dopamine COVERED particles (not recovered, several instances in the manuscript)
- Line 86: “proper size” what is the proper size, and why is it proper?
- Reword paragraph lines 88-93. The sentences are too long, and probably the order of thoughts should be changed.
- Lines 94/95 and lines 104/105: grammar: use past tense and passive, i.e. A qualitative change…..was observed.
- Table 1
- Round the values sensibly in all rows. E.g. Elongations at break of 132.19% don’t make any sense in view of the error of the method
- PD-BaSO4 (wt.%)
- Line 104 and several subsequent passages: Don’t use “increase in mechanical properties” or the like – be specific, e.g. increase in Young’s modulus and stress at break.
- Line 112: of more particle agglomerates (or aggregates)
- Line 118: …and adhesion. Therefore, …
- Line 126-128: reword
- Line 136: …counterpart. The compression…
- Line 151: remove the dot first in line
- Figure 6: It would be helpful to (also here? Alternatively? In the S.I. ??) have a depiction with % release
- Materials and Methods: always state (City, Country) not only country of the provider
- Headline 4.3 3D printing (not 3.D printing)
Author Response
Response to Reviewer 2 Comments
First of all, thanks to the reviewers for helping us improve our paper with their constructive comments. We have carefully considered all of them and the changes have been performed accordingly.
The comments of the reviewers are answered individually and specifically (in bold)
The authors describe a composite system based on a PLA matrix with incorporated polydopamine-coated BaSO4 particles. By combining these structural features, several points are addressed that are of relevance for application of polymers as bone implants. The benefits include: radiopacity, increased toughness while increasing the elongation at break, and therefore less brittle materials, and the ability of increasing drug loading and controlling the release.
While the article is allover well written, I have two major concerns, a minor concern, and several small remarks.
- My main concern 1 is the choice of PLA as matrix material. While there are probably hundreds of articles regarding the use of PLA as implants and drug releasing devices, especially for bone applications PLA and derivatives are questionable as major component. Since a long time it is known that while such PLA (and related polyesters) implant materials may help in the initial phase of bone regeneration, the subsequent acidification of the bone tissue because of material degradation may lead to problems in the clinic. See for example J. Biomed. Mater. Res. 1997, 38: 105-114 or also Biomaterials 2004, 25, 5735. This is why generally these days not major bone replacements are made from such materials, but may be small additions such as screws or drug releasing particles, or extra measures are undertaken to make sure that the acidification is counteracted by coatings, additions etc. So, ideally for this study, a different polymer matrix should have been chosen. As this is a bit late now, the authors should address this point with relevant literature, perhaps defining the exact application of their potential implant material. Also, it could be discussed whether the BaSO4 or PD may positively contribute, though their influence will be small due to the low content. In the end, the potential impact of the work is hampered by the material selection.´
Thanks for the appointment of the problems in the surgery with high size replacement of bones by PLA. In fact, there are a lot of references of the use of PLA as bone implant. In our case the application is focused in the improvement in mechanical properties and bioactivity of the bone fixation devices of small size, that having also BaSO4 in the formulations, will not a priori present relevant problems because of acidification of the medium because of by-products of biodegradation. The acidification of PLA by-products is out of scope of this work even though the BaSO4 particles could help in resolving issues related to it providing a more basic degradation by products in polyester biodegradation. This subject is under study by us and we will treat it in a posterior work.
Related to the comment the following sentences are incorporated in the introduction section
“[…] Having a glass transition temperature around 60 ºC, at body temperature semicrystalline PLA will be in glassy state (vs. elastomeric) and being crystalline at the same it will fulfil the requirements of a high stiffness and strength that a polymeric biomaterial requires for bone reconstruction in the form of fixation devices of small size [2] […]”
“[…] The aim of this work is to study a novel polymeric biodegradable composite system having potential for use as a biodegradable polymeric fixation template applicable to bone reconstruction. […]”
- Major concern 2: The composites are formed at 200 °C. If you have ever performed a DSC and/or TGA of PLA, you will have noticed that at this temperature the degradation of PLA already starts. It is necessary to provide data of the PLA before/after composite formation (NMR, GPC) to show that the PLA did not degrade and that you actually have the molar mass that you claim.
We have included the GPC in the supporting information, as you can see the matrix is not degraded, also the scaffolds do not show degradation problems because they were printed in solution.
In row 278 the following text has been added:
“Gel permeation chromatography (GPC) tests were carried out before and after blending process to check that the matrix is not degraded (see supporting information S8).”
- Minor concern: If you already have specified that your materials shall be implanted in bone, why have you studied the influence on fibroblasts, rather than performing experiments with osteoblasts (and perhaps also osteoclasts)? True, very often fare used for basic toxicity tests, but such other work would increase the validity of the work a lot.
Thanks again for the appointment of a more appropriate analysis, we agree with the reviewer. We include the following words to the text: ‘… as a basic toxicity test’.
Small remarks
- Line 4-42: The sentence should be split in two after ref. [3,5]
Done
- Line 82. Again, start a new sentence after …system [3,5]. The yield…
The paragraph containing this sentence has been rewritten.
- Dopamine COVERED particles (not recovered, several instances in the manuscript)
Done
- Line 88: “proper size” what is the proper size, and why is it proper?
It has been changed the sentence for a better understanding.
“As identified in our previous works with PLA/BaSO4 composites [3,5], the plastic deformation in composites occurs because the rigid particles act as stress concentrators and, after particles debonding from the matrix and being of a 0.7-1.9 µm size [27], activate, at a point of applied stress, crazing and shear yielding deformation mechanisms. Since polydopamine covered particles (PD-BaSO4) present 1.25 µm size and interactions with the matrix through the polydopamine interface, in these composites, increases in all mechanical properties (elastic modulus, strength, ductility and toughness) are achieved [28].”
- Reword paragraph lines 88-93. The sentences are too long, and probably the order of thoughts should be changed.
The description of results and discussion of Table 1 has been rewritten correcting the sentences and clarifying it.
“Table 1 shows the tensile mechanical properties of PLA/PD-BaSO4 composites. As can be observed, up to a 2 wt. % BaSO4 content in composites, Young’s modulus, ductility and toughness improvements can be observed. Further, if these results are to be compared with those of PLA/BaSO4, the non PD functionalized composite counterparts, enhancements in stiffness and strength together with ductility and toughness can be noticed in the former. Please also note that a huge > 2300 % increase in elongation at break are determined in these novel composite formulations in regard to neat PLA, which brings about a dramatic improvement in toughness. This is attributable to specific interactions stablished between the ester groups of polylactide and the alcohol groups of polydopamine coating of the particles surface [18,29] bringing about a stronger fiber/matrix interface. It is also noticeable that a 30% increase in the elongation at break is obtained in PD coated BaSO4 PLA composites in regard to the composites without polydopamine coating [3]. Finally, beyond the 2 wt. % particle composition, a slight general decrease in mechanical properties are noticeable, suggesting the existence of particle aggregates and bundles beyond this point.”
Lines 94/95 and lines 104/105: grammar: use past tense and passive, i.e. A qualitative change…..was observed.
These sentences have been combined with the previous paragraph (See above text).
Table 1
- Round the values sensibly in all rows. E.g. Elongations at break of 132.19% don’t make any sense in view of the error of the method
Done, values have been rounded using a decimal.
- PD-BaSO4 (wt.%)
Done
- Line 104 and several subsequent passages: Don’t use “increase in mechanical properties” or the like – be specific, e.g. increase in Young’s modulus and stress at break.
Done
- Line 112: of more particle agglomerates (or aggregates)
Changed to aggregates
- Line 118: …and adhesion. Therefore, …
Done
- Line 126-128: reword
This is the proposed text:
“In this case, mechanical properties were measured in compression to mimic the working conditions of the device. Non-reinforced polylactide (as reference) and the 10 wt.% PLA/PD-BaSO4 scaffolds with 55 % of porosity have been tested.”
Line 136: …counterpart. The compression…
Done
- Line 151: remove the dot first in line
Done
- Figure 6: It would be helpful to (also here? Alternatively? In the S.I. ??) have a depiction with % release
We included it in the SI section and add the following text in row 223:
[…](see supporting information S7).[…]
- Materials and Methods: always state (City, Country) not only country of the provider
Done
- Headline 4.3 3D printing (not 3.D printing)
Done
Reviewer 3 Report
The manuscript: “Benefits of polydopamine as particle/matrix interface 2 in polylactide/PD-BaSO4 scaffolds” by Sadaba et al. reports the versatility of polydopamine when applied as a particle coating in a composite of polylactide. The obtained particles provide new prospective for the development of novel biomaterial formulations manufactured by 3D printing technologies. The study is very interesting, well written but I think it can be considered for publication after a major revision.
Please take into consideration the following remarks:
- First of all, where is the supporting information (S1-S6)?
- Why didn't you use abbreviations, especially since you also have the abbreviations section? PD, instated of polydopamine throughout the text, and PLA instated of polylactide? Please be careful to this aspect.
- Line 67 – Add the full word for MWCNT and PU.
- You have explained that the release of levofloxacin was performed at pH 5 to simulate the state of infection and at body temperature of 37 °C, but you haven’t explain anywhere, why have you chosen pH 10 (trizma buffer) to submerge the scaffolds for absorption/release test.
- Line 151-165: I think it will be useful to synthesize the data about metabolic activity of HDFs in a table.
- You haven’t mentioned details about PD in section 4.1. Materials.
- Line 279 – Please add italic style: 4.3. 3D printing of radiopaque scaffolds
- Add all the abbreviations in the specific section: e.g. Scanning Electron Microscopy (SEM), Transmission Electron Microscopy (TEM), Computer Aided Drawing (CAD), Human dermal fibroblasts (HDFs) and so on.
Author Response
Response to Reviewer 3 Comments
First of all, thanks to the reviewers for helping us improve our paper with their constructive comments. We have carefully considered all of them and the changes have been performed accordingly.
The comments of the reviewers are answered individually and specifically (in bold)
The manuscript: “Benefits of polydopamine as particle/matrix interface in polylactide/PD-BaSO4 scaffolds” by Sadaba et al. reports the versatility of polydopamine when applied as a particle coating in a composite of polylactide. The obtained particles provide new prospective for the development of novel biomaterial formulations manufactured by 3D printing technologies. The study is very interesting, well written but I think it can be considered for publication after a major revision.
Please take into consideration the following remarks:
- First of all, where is the supporting information (S1-S6)?
Some problem may have occurred with the supporting information, since we uploaded it together with the article, we will include it with the revised version hopefully it will be visible.
- Why didn't you use abbreviations, especially since you also have the abbreviations section? PD, instated of polydopamine throughout the text, and PLA instated of polylactide? Please be careful to this aspect.
Done
- Line 67 – Add the full word for MWCNT and PU.
Done
- You have explained that the release of levofloxacin was performed at pH 5 to simulate the state of infection and at body temperature of 37 °C, but you haven’t explain anywhere, why have you chosen pH 10 (trizma buffer) to submerge the scaffolds for absorption/release test.
The sentence has been completed with the following words: “[…] for facilitate the adsorption”
- Line 151-165: I think it will be useful to synthesize the data about metabolic activity of HDFs in a table.
These data are included in S.I. and added in row161 the following text:
[…](see also Supporting Information S5). […]
You haven’t mentioned details about PD in section 4.1. Materials.
As PD results from a spontaneous polymerization of dopamine chloride, the following sentence is provided in Section 3.1. Materials I reference to it is provided “Coating of particles covered with polydopamine (PD) to obtain PD-BaSO4 were obtained as in our previous article using basic pH 8.5 for 24 hours followed by filtration and drying in a vacuum oven overnight [17].”
- Line 279 – Please add italic style: 3. 3D printing of radiopaque scaffolds
Done
- Add all the abbreviations in the specific section: e.g. Scanning Electron Microscopy (SEM), Transmission Electron Microscopy (TEM), Computer Aided Drawing (CAD),
Human dermal fibroblasts (HDFs) and so on.
Done
Round 2
Reviewer 1 Report
No comments.
Author Response
Thank you for the supervision
Reviewer 2 Report
The authors improved the manuscript in the revision that is now basically in an acceptable form from my perspective.
However, the GPC curves of PLA before / after composite formation are similar though not identical. As no numbers are given, it is hard to judge whether there was degradation (probably to a minor extent).
I therefore recommend to include a statement in the main text, e.g. as first statement in the results and discussion part along the lines of "Composite formation was performed by XXX. GPC analysis of PLA before and after the composite formation gave molar masses of XXX and YYY with PDIs of WWW and ZZZ. In view of the methodological error these values are different, so some degradation was taking place due to the heating /// these values are not significantly different, demonstrating that the composite formation did not impact the integrity of the polymer.
Author Response
The following text has been added to the lines 281-284:
The weight average molecular mass (Mw) and dispersity (D) of PLA pellets before processing were respectively Mw=183758 gmol-1 and D=1.8 whereas, after processing in a PLA/BaSO4 composite, the values obtained for PLA were Mw=174980 gmol-1 and I=1.8.
Reviewer 3 Report
The authors have made significant effort to address all my comments and concerns. I sincerely recommend the publication of the manuscript.
Author Response
Thank you for the suggestions